# Opinion: Cloud-phase climate feedback and the importance of ice-nucleating particles

Benjamin J. Murray[1], Kenneth S. Carslaw[1], Paul R. Field[1,2]

[1]Institute for Climate and Atmospheric Science, School of Earth and Environment, University of Leeds, LS2 9JT Leeds, United Kingdom

[2]Met Office, Exeter EX1 3PB, United Kingdom

*Correspondence to*: Benjamin J. Murray (b.j.murray@leeds.ac.uk)

**Abstract.** Shallow clouds covering vast areas of the world's mid- and high-latitude oceans play a key role in dampening the global temperature rise associated with $CO_2$. These clouds, which contain both ice and supercooled water, respond to a warming world by transitioning to a state with more liquid water and a greater albedo, resulting in a negative 'cloud-phase' climate feedback component. Here we argue that the magnitude of the negative cloud-phase feedback component depends on the amount and nature of the small fraction of aerosol particles that can nucleate ice crystals. We propose that a concerted research effort is required to reduce substantial and important uncertainties related to the poorly understood sources, concentration, seasonal cycles and nature of these ice-nucleating particles (INPs) and their rudimentary treatment in climate models. The topic is important because many climate models may have overestimated the magnitude of the cloud-phase feedback, and those with better representation of shallow oceanic clouds predict a substantially larger climate warming. We make the case that understanding the present-day INP population in shallow clouds in the cold-sector of cyclone systems is particularly critical for defining present-day cloud phase and therefore how the clouds respond to warming. We also need to develop a predictive capability for future INP emissions and sinks in a warmer world with less ice and snow and potentially stronger INP sources.

## 1 Introduction

Projections of global warming due to increased anthropogenic greenhouse gas concentrations is of central importance for our society. We need these projections to be sufficiently accurate in order to effectively plan adaptation and mitigation strategies and also to provide a robust basis for plans to curb carbon emissions. However, substantial and poorly defined uncertainties exist in our climate models. While it is unambiguous that greenhouse gas emissions are leading to a warmer climate, our climate models are hugely divergent in how much the world will warm in future (see Figure 1 and Box 1). Apart from the obvious societal benefits of reducing uncertainty, improvements to climate predictions are estimated to carry a multi-trillion dollar value (Hope, 2015).

The way that clouds respond to forcing by $CO_2$ in models is one of the key sources of uncertainty in climate projections. These feedbacks on climate can either dampen (negative feedback) or amplify (positive feedback) climate warming. In fact, some models predict a negative overall cloud feedback, whereas others predict an overall positive feedback (Figure 1). The divergence of the treatment of clouds amongst models correlates with the predicted amounts of warming for a doubling of $CO_2$ (known as equilibrium climate sensitivity, ECS, see Box 1), with negative feedbacks resulting in smaller ECS values and vice versa.

There has been a shift amongst some more recent models to larger ECS values. In the 2013 IPCC assessment the estimates of ECS ranged from 1.5 to 4.5°C (Collins et al., 2014), whereas 10 out of 27 models that will inform the next IPCC assessment, have ECS values greater than 4.5°C (Zelinka et al., 2020). Whether these high ECS values (Gettelman et al., 2019; Zelinka et al., 2020; Bodas-Salcedo et al., 2019) are possible or probable is a matter of debate (Palmer, 2020; Forster et al., 2020; Sherwood et al., 2020). Nevertheless, one of the key differences between the older CMIP5 (Coupled Model Intercomparison Project) models and the new CMIP6 models is the treatment of clouds in the mid- to high-latitudes where clouds can persist in a mixed-phase state. Many of the CMIP6 models have a much more positive cloud feedback at latitudes poleward of 45°, which correlates with higher ECS values (Figure 1b). This illustrates the key role that clouds, particularly shallow marine clouds in the mid- and high-latitudes, play in inter-model variations in ECS (Ceppi et al., 2017; Zelinka et al., 2020; Gettelman et al., 2019; Andrews et al., 2019). We argue that this issue has to be addressed urgently.

Liquid-only clouds in the marine boundary layer at low-latitudes are generally expected to decrease in amount in a warmer world, exerting a positive feedback (Ceppi et al., 2017). However, for clouds at higher latitudes or greater altitudes where the temperature is below the freezing point of water, the response to warming can be entirely different (see Figure 2). The key difference in 'mixed-phase' clouds is that the formation and precipitation of ice crystals can strongly reduce the amount of supercooled liquid water, which accounts for most of the cloud reflectivity. If aerosol particles capable of nucleating ice, ice-nucleating particles (INPs), are present and are active at the local cloud temperature, then the supercooled liquid water content and albedo of these clouds can be dramatically reduced through ice-related microphysical processes (Vergara-Temprado et al., 2018; Komurcu et al., 2014; Storelvmo, 2017). In a warmer future climate, water will replace ice and therefore the cloud will

have a greater albedo. For clouds over dark surfaces such as oceans, the cloud-phase feedback caused by this simple thermodynamic change is negative, but its magnitude is highly uncertain (Storelvmo, 2017; Storelvmo et al., 2015; Tan et al., 2016; Frey and Kay, 2018).

Here we argue that although temperature changes are the primary driver of changes in ice formation, the magnitude of the cloud-phase feedback is directly related to the spatial and temporal distribution of the atmospheric INP population and also how this INP population may change in the future. While we have learnt a great deal from recent field and laboratory work about INPs in mid- to high-latitudes (~45-70°), the region critical for the cloud-phase feedback, we need a much better understanding of sources and sinks of INP as well as the nature of INPs in both hemispheres. We finish by outlining what research needs to be undertaken to reduce the uncertainty associated with the cloud-phase feedback.

## 2. The cloud-phase feedback and the importance of ice-nucleating particles

The first description of the cloud-phase feedback in the literature was over 30 years ago by Mitchell et al. (1989). They found that on including a treatment of cloud phase in their model the global mean temperature change on a doubling of $CO_2$ decreased from 5.2 to 2.7°C. This and more recent work point to a strong but highly uncertain negative feedback focused in the mid- and lower high-latitudes (Storelvmo et al., 2015; Tan et al., 2016; Ceppi et al., 2017; Ceppi et al., 2016; McCoy et al., 2018; Frey and Kay, 2018). The divergent representation of cloud feedbacks at these latitudes leads to huge variability in mid- to high-latitude cloud feedback (-0.63 to +0.68 in CMIP6 models) and a strong positive correlation with ECS (see Figure 1b). In climate models, which probably do not represent all the key processes, these uncertainties in feedbacks stem from what assumptions are made about the existence and radiative properties of mixed-phase cloud.

The core physical process that drives the cloud-phase feedback is the transition to clouds with more liquid water and less ice as the isotherms shift upwards in a warmer world (see Figure 2). The short wave (SW) cloud radiative effect (CRE) of clouds is strongly dependent on their liquid water content since liquid clouds tend to be made up of many cloud droplets of 10s of micrometres in diameter, which scatter shortwave radiation very effectively. In contrast, glaciation of a supercooled cloud results in far fewer particles of larger sizes and consequently shorter lifetimes which reflect much less sunlight. Hence, the microphysical processes that lead to glaciation and depletion of liquid water content are important for cloud feedbacks (Vergara-Temprado et al., 2018; Storelvmo et al., 2015).

The shift to fewer, but larger hydrometers when a supercooled cloud glaciates is a result of the abundance of aerosol available for nucleating cloud droplets and ice crystals, as well as the various ice-related microphysical processes which occur subsequent to ice nucleation. The aerosol particles that form cloud droplets, cloud condensation nuclei (CCN), are relatively common with 10s to 100s per $cm^3$ over the remote oceans (and much greater in air with continental influence). In contrast the concentration of INPs are typically many orders of magnitude smaller (DeMott et al., 2010; Kanji et al., 2017). Hence, a small subset of cloud droplets may contain INP (after serving as CCN themselves) and if these droplets are sufficiently cold, they

will freeze (Koop and Mahowald, 2013). These frozen droplets then find themselves in an environment that is strongly supersaturated with respect to ice (~10% in a liquid cloud at -10°C), hence they grow rapidly. Within minutes they reach 100s of micrometres in diameter, depleting liquid water through diffusional growth (Wegener–Bergeron–Findeisen process, WBF) and accretion of droplets (riming) as they grow and precipitate. In some situations the impact of INP will be amplified through secondary ice production (SIP) where a range of mechanisms are thought to result in the production of additional ice crystals (Field et al., 2017). It should be borne in mind that these processes (SIP, WBF, riming) subsequent to ice nucleation are also relatively poorly understood and also need attention (Komurcu et al., 2014). However, primary ice production initiates these subsequent ice-related processes, therefore the role of INPs in the cloud-phase feedback is the focus of this paper. Modelling work suggests that at concentrations of ice crystals above about 1 $L^{-1}$ there are dramatic reductions in liquid water, but smaller concentrations also deplete the liquid water path and reduce albedo (Vergara-Temprado et al., 2018; Stevens et al., 2018). However, the relationship between INP concentration and cloud glaciation is complex and governed by the WBF process (Desai et al., 2019). In some publications, CCN and INP are collectively referred to as 'cloud-forming nuclei'. In fact, for INP, the opposite is true: they should be regarded as cloud (or at least albedo)-destroying agents in shallow supercooled clouds.

The underpinning physical principles of the cloud-phase feedback are illustrated in Figure 2 and 3. Generally, a warmer world results in a larger proportion of the marine boundary layer containing clouds at temperatures which do not support ice formation and growth. The greater prevalence of reflective droplets in these warmer clouds combined with less precipitation leads to less shortwave radiation being absorbed by the ocean and a negative climate feedback. The strength of this feedback depends on the balance between ice and supercooled water in the present and future climate (Figure 3); however the cloud-phase feedback is treated in climate models with varying levels of detail.

We hypothesise three ways in which the nature and concentration of INPs can directly modulate the strength of the feedback (Figure 4). Firstly, the more ice in clouds in the present climate, the stronger the negative cloud-phase feedback, while in clouds which are mainly composed of supercooled water the cloud-phase feedback will be relatively weak (Figure 4a). Since the amount of ice in many shallow clouds is strongly influenced by the INP population, there are likely to be regional and seasonal variations in the cloud-phase feedback. If our understanding is correct, then regions with strong INP sources should have a more negative cloud feedback than regions with weaker INP sources (also see Figure 3a and b). However, at present we have insufficient measurement and modelling data to test this hypothesis.

Secondly, the magnitude of the cloud-phase feedback will depend on the nature of INP because different types of INP have very different temperature dependencies, and this directly affects how the mixed-phase part of the cloud responds to warming (Figure 4b). The increase in INP concentration, and hence ice particle formation, per degree of cooling is greater for a material with a steep slope, such as mineral dust (Atkinson et al., 2013; Harrison et al., 2019), than a material with a shallower slope, such as fertile soil dust (Steinke et al., 2016; O'Sullivan et al., 2014). In the case of a steep slope, a warming climate will cause a greater reduction in the concentrations of INPs active at cloud temperatures than in the case of a shallow slope. Hence, there

will be a stronger feedback in the case of a steep slope. However, the temperature dependence of INP from different sources relevant for the cloud-phase feedback is poorly understood, and our understanding of how clouds respond to variations in the nature of INP is far from complete.

Thirdly, INP sources, processing and removal in the atmosphere are also likely to change with a changing climate (Figure 4c). For example, it has been suggested that less snow and ice cover may lead to more widespread emission sources and higher dust emissions rates at high latitudes (Tobo et al., 2019; Prospero et al., 2012; Sanchez-Marroquin et al., 2020; Amino et al., 2020) (we discuss this further in section 6). Also, INP emissions have been linked to environmental factors such as rain fall, hence a warmer wetter world may lead to enhanced INP emission rates from some terrestrial sources (Conen et al., 2017;
Huffman et al., 2014; Hara et al., 2016). Higher INP concentrations would lead to more ice in cold clouds, which would lead to a positive feedback (see Figure 3c). But, it is also conceivable that INP sources might weaken if, for example, dust sources become vegetated. Alternatively, loss mechanisms might be enhanced in a warmer world with more precipitation. This would lead to a stronger negative feedback. Furthermore, biological processes which result in very active biogenic INP (primary biological particles, by-product fragments and macromolecules) (Hill et al., 2016; O'Sullivan et al., 2015), may also respond
to a changing climate. Hence, a correct representation of INP and a link to the type of aerosol and the sources is necessary to represent this aspect of the cloud-phase feedback process.

It has become clear over the last few years that many models may overestimate the magnitude of the cloud-phase feedback, especially in the Southern Ocean. There are well-known model biases in the Southern Ocean with too much SW radiation making it to the surface due to shallow clouds not being sufficiently reflective (Bodas-Salcedo et al., 2012; Trenberth and
Fasullo, 2010). In many models, these shallow clouds contain too little supercooled water, exposing the dark ocean underneath and resulting in sea surface temperatures around 2°C too warm (Wang et al., 2014). This bias has profound implications for the strength of the cloud-phase feedback. Tan et al. (2016) demonstrated that the strength of the cloud-phase feedback was strongly dependent on the amount of supercooled liquid water in present-day clouds (SI Figure 1). The ECS in their control case, where the model was run in its default configuration was 4.0°C, whereas when the amount of supercooled water in the
present day climate was increased to be more consistent with satellite data the ECS increased to 5.3°C. Similarly, Frey and Kay (2018) showed that ECS increased from 4.1 to 5.6 when they increased the amount of supercooled water to better match observations of absorbed shortwave radiation over the Southern Ocean. The fact that ECS is sensitive to the balance between supercooled water and ice in clouds means that we have to improve our understanding of ice-related microphysical processes. In particular, we need a concerted effort to understand the atmospheric abundance of INPs, the aerosol type which catalyses
ice formation in mixed phase clouds and plays a major role in defining the cloud-phase feedback.

### 3. To what extent is the persistence of supercooled liquid clouds related to ice nucleation?

In the absence of collisions with ice crystals, water droplets can freeze both homogeneously, i.e. spontaneously, or heterogeneously, where an impurity catalyses freezing. Homogeneous nucleation defines the lower limit to which supercooled clouds can persist in the absence of INP. The exact temperature limit depends on dynamics and microphysics, but homogeneous
nucleation becomes increasingly important below about -33°C (Herbert et al., 2015; Koop and Murray, 2016) which is consistent with the lack of supercooled water in shallow clouds below about -35°C (Kanitz et al., 2011; Morrison et al., 2011; Hu et al., 2010) (also SI Figure 2).

There are many aerosol particle types that possess the capability to nucleate ice, from mineral dusts to biological particles and combustion aerosol to fertile soil dusts (see the reviews of Kanji et al. (2017), Murray et al. (2012) and Hoose and Möhler
(2012)). One of the striking and important aspects of INPs is that particles with the capacity to serve as immersion mode INPs are rare in comparison to those capable of serving as cloud condensation nuclei. Even within a specific category of INPs, not all particles with a particular composition will nucleate ice. For example, ice nucleation by desert dust is thought to depend on the presence of K-feldspar (Harrison et al., 2019; Atkinson et al., 2013; Peckhaus et al., 2016) and even then only a fraction of K-feldspar grains possess active sites capable of nucleating ice at around a particular characteristic freezing temperature
(Holden et al., 2019). The fact that ice nucleation, at least on some materials, is a site-driven process means that it is not possible to define the ice-nucleating ability of an aerosol population using macroscopic properties in a manner that is analogous to droplet formation on soluble particles, which depends solely on the bulk chemical composition. Hence, we have to empirically quantify the ability of specific particle types by describing the distribution of sites across the particle population using quantities such as the INP concentration spectrum or the active site density spectrum.

In general, the INP concentrations in air masses associated with land are higher than those with a strong marine influence (McCluskey et al., 2018b; Vergara-Temprado et al., 2017; DeMott et al., 2016; Welti et al., 2020). This terrestrial-marine divide is related to the sources in the two environments. There is clearly a source of highly active INP in sea water (Wilson et al., 2015; Schnell and Vali, 1975; Irish et al., 2019b), but the sea spray production process only produces rather low INP concentrations (Vergara-Temprado et al., 2017; McCluskey et al., 2018a; DeMott et al., 2016). In contrast, there are a plethora
of potential INP sources on land including mineral dusts, biogenic particles and combustion particles (Kanji et al., 2017; Murray et al., 2012).

This divide between terrestrially influenced regions and remote oceans is reflected in the extent to which shallow clouds supercool. For example, satellite data indicates that liquid clouds over the Southern Ocean supercool extensively, whereas clouds over Europe, where there are stronger INP sources, supercool much less (Choi et al., 2010; Storelvmo et al., 2015;
Kanitz et al., 2011; Hu et al., 2010). Furthermore, it has been shown that the degree of supercooling correlates with the presence of specific aerosol species such as mineral dust (Tan et al., 2014; Choi et al., 2010). Also, it has been shown using satellite data that there is a large contrast in the contribution of cloud phase changes to changes in cloud optical depth with temperature

between land and ocean, which points to the importance of INP (Tan et al., 2019). Hence, there is a clear link between the degree of supercooling and aerosol type, which needs to be represented routinely in climate models.

## 4. How well do models represent phase partitioning in climate models?

Current models are hugely divergent in their representation of the amounts of supercooled water (Komurcu et al., 2014; Zelinka et al., 2020; McCoy et al., 2015; McCoy et al., 2018; Cesana et al., 2015). For example, in an intercomparison of cloud water between several climate models Komurcu et al. (2014) found that in some models liquid water had largely been removed as warm as -10°C, while in other models unrealistically high amounts of liquid water persisted down to -35°C. Some of these models also deviate strongly from satellite measurements of cloud-top phase (Cesana et al., 2015; Komurcu et al., 2014).

The reasons for the model discrepancies are complex. Cesana et al. (2015) conclude that models with more complex microphysics tend to have a better representation of ice phase. Also, Komurcu et al. (2014) conclude that the inter-model variability they report was related in part to the specifics of the ice nucleation scheme, but also to the representation of other ice-related microphysical process. However, it is important to bear in mind that many of the relevant processes occur on scales finer than the grid resolution of climate models, and parametrizations of these processes can affect the distribution and amounts of ice and liquid (Kay et al., 2016). Nevertheless, the amount of supercooled liquid water in climate models is highly sensitive to the treatment of primary ice production (Vergara-Temprado et al., 2018). Overall, the representation of phase partitioning in models is massively divergent and this likely contributes to the variable cloud feedbacks and ECS values (Bodas-Salcedo et al., 2019). In the future, models need to improve their representation of ice-related microphysical processes, in particular, they need to include a direct link to aerosol type, specifically INP, in order to improve the representation of clouds phase and the response of clouds to a warming world.

## 5. What are the meteorological conditions most important for the cloud-phase feedback?

Detailed analysis of model biases over the Southern Ocean have shown that the cold air-outbreaks (CAOs) are of central importance to the cloud-phase feedback (Bodas-Salcedo et al., 2016; Bodas-Salcedo et al., 2014). Marine COAs are high impact events where cold air flows from higher latitudes over a warmer ocean (SI Figure 3). This creates the conditions for shallow supercooled cloud systems as heat, moisture and aerosol is mixed into cold air. The strongest CAOs are associated with the cold sector of extratropical cyclone systems which tend to draw air from the polar or cold continental regions (Fletcher et al., 2016; Pithan et al., 2018).

Modelling work has shown that CAO cloud systems are strongly impacted by INP, with low INP concentrations leading to more extensive highly reflective stratus clouds whereas high INP concentrations tends to lead to much patchier convective cloud with local albedos many 100s W m$^{-2}$ lower (Vergara-Temprado et al., 2018). This is illustrated in Figure 5 where a

cyclone system was simulated by nesting a high resolution (7 km) region within a global model (Vergara-Temprado et al., 2018). Two cases are shown in Figure 5, one with INP concentrations representative of the terrestrial mid-latitudes (high [INP]) and one representative of the Southern Ocean (low [INP]). The mean cloud reflectivity in the cold sector is lower by 100s W m$^{-2}$ in the high [INP] case relative to the low [INP] case, and Vergara-Temprado et al. (2018) shows that the reflected shortwave flux increases with increasing INP concentration. This illustrates that correctly representing primary ice production, and INP, is critical for maintaining the amount of supercooled water in clouds and their albedo. More importantly, although various processes in models could be adjusted to match present-day measurements, this would not address how INP influences the response of the clouds to warming.

## 6. What do we currently know about atmospheric INP in the regions important for the cloud-phase feedback?

Our knowledge of the global distribution, seasonal cycle and sources of these enigmatic particles is in its infancy. However, we argue that the documented importance of CAO clouds (Bodas-Salcedo et al., 2016), allows us to focus on understanding aerosol and INP sources in these specific environments. The air flow in CAOs is well defined with air streaming out of the colder high latitudes, into the mid-latitudes. These cloud systems are therefore impacted by i) high latitude aerosol and terrestrial biogenic INP sources; ii) sea spray which carries biogenic INP; iii) and INP in the free troposphere from more distant sources entrained into the boundary layer. Hence, mid and high-latitude sources of INP may have a disproportionate effect on climate through their influence on shallow clouds.

What do we know about INP at mid-to high-latitudes, and specifically in environments that have the potential to directly impact CAOs? Measurements of INP concentrations in regions which may impact CAOs are summarised in Figure 6. It is striking how variable INP concentrations are, both in space and time. If we take 1 INP L$^{-1}$ as a reference value, where ice formation is thought to substantially reduces the liquid water path and albedo, then this threshold is reached anywhere from around -10°C to temperatures where we expect homogeneous freezing to dominate primary ice production (<-35°C). This temperature range is not an uncertainty, but rather a range of atmospheric states that we need to understand because it is relevant to present-day mixed-phase clouds and future feedbacks.

## 6.1 INP in the northern mid- to high-latitudes

The limited INP concentration data in Figure 6 indicate that the INP concentrations in the northern hemisphere are generally higher than in the southern hemisphere. This may be related to the proximity of terrestrial sources in the northern hemisphere that are less common in the southern hemisphere. Over recent years it has become increasingly apparent that there are significant dust emissions from a plethora of high-latitude sources, such as pro-glacial deposits (Bullard et al., 2016; Prospero et al., 2012). Samples from a handful of these sources have been shown to nucleate ice (Tobo et al., 2019; Paramonov et al., 2018; Sanchez-Marroquin et al., 2020) and dust from Iceland's deserts has been shown to be an important INP type across the

N. Atlantic and low Arctic (Sanchez-Marroquin et al., 2020). Further evidence for a strong terrestrial source of INP in the Arctic was found by Irish et al. (2019a) who found a correlation between INP concentrations and the time that air spent over bare land during late summer. In addition to mineral dust, which tends to control the INP population only below about -15°C (Murray et al., 2012), there is evidence that there are strong sources of terrestrial biogenic material active at much warmer temperatures across the Arctic (Tobo et al., 2019; Wex et al., 2019). Terrestrial biogenic material might be associated with sediments from rivers (Tobo et al., 2019) or vegetated areas (Conen et al., 2016; Schnell and Vali, 1976). In fact, it has been suggested that biogenic ice-nucleating material may account for the INP active at the highest temperatures in Figure 6 (Wex et al., 2019).

In addition to terrestrial sources, there are multiple studies showing that there is a biogenic source of INP in sea water which can become aerosolised through the action of waves and subsequent bubble bursting (Schnell, 1977; Schnell and Vali, 1975; Wilson et al., 2015; Irish et al., 2019b; DeMott et al., 2016; Irish et al., 2017; Creamean et al., 2019). Modelling (Vergara-Temprado et al., 2017) and measurements (McCluskey et al., 2018b; McCluskey et al., 2018a) suggests that this source produces sea spray aerosol which are relatively ineffective INP, with activities orders of magnitude (on a per surface area basis) than mineral dust. Marine biogenic INP may define a baseline INP concentration in environments which lack other more active INP types (McCluskey et al., 2018a; Vergara-Temprado et al., 2017; Schill et al., 2020), and it is conceivably an important source in windy CAOs. In the northern hemisphere, even the small quantities of dust transported from the low latitude source regions may dominate over marine sources of INP for much of the time (Vergara-Temprado et al., 2017) and local terrestrial sources may episodically swamp both marine and low latitude sources (Sanchez-Marroquin et al., 2020). Taking all this together, the INP population in the northern hemisphere high latitudes appears to be a complex mixture of different INP types from the marine and terrestrial environment.

The observed strong seasonal dependence of high-latitude northern hemisphere INP concentrations could give us a clue to how INP might change with climate (our third hypothesis, Figure 4c). These dependencies are clearest in the multi-season data presented by Wex et al. (2019) for four locations around the Arctic. The highest INP concentrations occur in the spring, summer and autumn when high latitude marine and terrestrial sources become ice free and when biological activity is at its maximum. The implications of these data are that there is a local biogenic source of INP in the northern high latitudes and that as the ice and snow season shortens with a warmer climate, these sources may be active for more of the year and more sources may become available, which would positively feedback on climate through increased ice production in clouds (Figure 3c). In addition to this, it has been argued that high latitude dust sources associated with glaciers will become more active in the future (Bullard et al., 2016) and it was recently shown that mineral dust emissions from the coastal areas of Greenland have increased in the last few decades (Amino et al., 2020). However, paleo records indicate that warmer periods are generally less dusty than dry periods, although this may reflect a combination of lower latitude sources being more active and increased transport to high latitudes during glacial periods (Lamy et al., 2014; Fischer et al., 2007). Hence, it may be that the glacial-interglacial trends in high latitude dust sources relevant for CAOs are decoupled from low latitude dust sources and the general dust loading

of the atmosphere. More work in identifying the sources of INP in the high latitudes and how these sources will respond to a changing climate is clearly required.

## 6.2 INP in the southern mid- to high-latitudes

It is unclear whether there are similarly strong INP sources in the southern hemisphere. Recent measurements over open ocean or in sea ice indicate that INP concentrations are generally very low (Schmale et al., 2019; McCluskey et al., 2018a); in fact,

these are amongst the lowest INP concentrations that have been measured anywhere on Earth. However, measurements at the coastal stations of McMurdo (Bigg and Hopwood, 1963) and Syowa (Kikuchi, 1971) indicate concentrations in excess of 1 L$^{-1}$ at -20°C. There are reports of dust uplift on the Antarctic peninsula (Bory et al., 2010; Asmi et al., 2018) and also in the dry McMurdo valleys (Lancaster, 2002). There are also dust sources more generally across the southern hemisphere, in particular dust from New Zealand and Patagonia are transported to the higher latitude Southern Ocean (Neff and Bertler, 2015) and dust

from Patagonia has been shown to be effective at nucleating ice (López et al., 2018). A significant input of INP to clouds in the Southern Ocean in the present climate would imply a strong negative cloud-phase feedback and that these clouds have a strong buffering effect on warming by anthropogenic $CO_2$. Conversely, if the INP source is weak, as contemporary measurements suggest (McCluskey et al., 2018a; Schmale et al., 2019), then the cloud-phase feedback would be far less negative than over the northern hemisphere. In addition, there is the potential that sources of INP in the southern hemisphere

become more prominent in the future as a response to warming, which would lead to a positive feedback. Clearly, more work needs to be done to assess sources, transport and nature of INP in both hemispheres.

## 7. Important areas of future research

The field of atmospheric ice nucleation and its role in defining the cloud-phase feedback is rapidly evolving. We have come a long way in recent years in defining the problem, improving our understanding of ice nucleation and building the capacity in

our models to deal with ice processes. However, while we can see that the climate system is very sensitive to the cloud-phase feedback, there are substantial knowledge, technology and skills gaps that need to be addressed in order to make quantitative predictions. Here we highlight some of the frontiers in the field which need to be addressed in order to reduce the uncertainty associated with the cloud-phase feedback.

*Control of primary ice production by INP in global climate models.*

Many global climate models do not represent the basic physical processes relevant for the cloud-phase feedback. For example, it has been shown that linking primary ice production to aerosol concentrations, amongst other changes, improved the representation of cold oceanic clouds (Gettelman et al., 2019; DeMott et al., 2010). This is an important result, but it must be acknowledged that there are many, sometimes more important, INP sources than low-latitude mineral dust, especially at high latitudes. Global climate models need to couple with a full model of INP, including sources and removal processes relevant to

specific cloud systems. Inclusion of INPs in climate models would open up the opportunity to simulate the number

concentration of primary ice particles, which is required for a realistic simulation of the chain of processes that control precipitation and cloud reflectivity (Vergara-Temprado et al., 2018).

*An INP measurement network.*

While aerosol properties such as their ability to activate to cloud droplets are made routinely around the world, INP concentrations are not. To improve the representation of the cloud-phase feedback we have to be able to represent INP concentrations in our models. This can only come from suitable measurements in the right places. We need a global network of INP measurement sites making year-round measurements across the full range of mixed-phase cloud conditions, with high priority in regions where CAOs are particularly important (i.e. ~45 to 70°).

*Instrument development.*

Until very recently, the INP measurement community has lacked instruments that can operate on an autonomous basis and can access the full range of INP concentrations and temperatures relevant for the cloud-phase feedback. In order to access the full range of INP concentrations, this will most likely require several separate instruments operating in parallel, targeting the full range of temperature and saturations over which clouds form in the atmosphere. Developments such as a new semi-autonomous portable expansion chamber INP counter (Möhler et al., 2020), the application of microfluidics technology (Tarn et al., 2020; Porter et al., 2020) and autonomous continuous flow diffusion chambers (Bi et al., 2019; Brunner and Kanji, 2020) may offer routes to much improved instrumentation for routinely quantifying INP concentrations.

*Quantifying INP sources and their physical, chemical and biological controls*. We have to understand quantitatively where INPs relevant for the cloud-phase feedback come from and what drives their emission. Sources in the Arctic appear to be strongly seasonal and are likely to respond to a changing climate. Sources in the southern hemisphere are even less well defined. Terrestrial high-latitude sources associated with pro-glacial deposits may be very important, but we are only just starting to quantify them.

*Dedicated field campaigns.*

We need field campaigns focused on quantifying the relationship between aerosol (INP and CCN), mixed-phase clouds and boundary layer dynamics. We need to understand how the processes in these cloud systems depend upon the sea surface temperature and changes in aerosol availability. As well as being key to the cloud-phase feedback, cloud systems in CAOs offer an opportunity to study a relatively repeatable weather regime that has a well-defined transition from mixed-phase stratus to shallow convective clouds.

*Development of global INP models which include all relevant sources.*

Many models create ice as a function of temperature but lack the link to aerosol; this has been shown to be inadequate (DeMott et al., 2010; Vergara-Temprado et al., 2018). We have begun to build models of the global distribution of atmospheric INP

(Vergara-Temprado et al., 2017; Hoose et al., 2010; Spracklen and Heald, 2014; Schill et al., 2020), but we currently lack an understanding of mid- and high-latitude sources. We must also represent the INP removal processes, which in turn depend on a correct representation of the microphysics. It is only with INP models where there is a link to surface properties in key source regions that we can expect to be able to predict how INP distributions will change in response to climate change.

*Cloud microphysics and dynamics.*

In addition to ice nucleation, other microphysical and dynamical processes are also extremely important for clouds and their response to a changing climate. Many of these other processes are also very uncertain, and are the topics of extensive review articles in themselves. For example, secondary ice production remains a major challenge and has the potential to amplify the effect of a small concentration of INP. However, even the basic mechanisms leading to ice multiplication are unclear (Field et

al., 2017; Korolev and Leisner, 2020).

## 8. Final comments

As a global civilisation striving to secure its future prosperity, wellbeing and sustainability, we need accurate predictions of our impact on Earth's climate. It is clear that our understanding of the cloud-phase feedback and ice-nucleating particles, as well as the representation of these processes in climate models, is limiting our ability to do this accurately. There is substantial

evidence that the cloud-phase feedback has been too negative in climate models and the correction of this will lead to larger ECS values. Whether these large ECS values are plausible is a topic of hot debate, but if they are not feasible then it seems some other feedback is (or feedbacks are) too positive. Nevertheless, it is becoming very clear that the cloud-phase feedback contributes substantially to the uncertainty in predictions of the rate at which our planet will warm in response to $CO_2$ emissions.

We argue that a concerted effort is needed from scientists working on different scales, from the detailed microphysical, biological and chemical processes associated with INP sources to those who can implement this knowledge to build a global understanding using state-of-the-art modelling tools. Without this underpinning knowledge and its suitable representation in our models, ECS will remain highly uncertain. But if it turns out that the larger ECS reported by some new climate models is correct, then society will need to act even more assertively to limit the accumulation of $CO_2$ in our atmosphere. Hence,

resolving the role of INP in the cloud-phase feedback needs to be a research priority for the coming years.

**Acknowledgments**

We thank Daniel McCoy, Mark Zelinka and Trudy Storelvmo for helpful discussions in relation to climate feedbacks in models. In particular, we thank Mark Zelinka for providing access to ECS and feedback data associated with Zelinka et al.

(2020). Conversations about historic data sets with Keith Bigg were invaluable. We are grateful to Heike Wex, Jessie Creamean, Jingwei Yun and Allan Bertram for sending us data used in Figure 6. We acknowledge the European Union Horizon 2020 (MarineIce, European Research Council, 648661 and PRIMAVERA 641727) and the Natural Environment Research Council (NERC, M-Phase NE/T00648X/1).

**Author contribution**

All authors contributed to the writing of this manuscript and the ideas expressed within it.

**Competing interests**

The Authors declare no competing interests

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

**Box 1: Climate change, Equilibrium Climate Sensitivity and Feedbacks**

The build-up of greenhouse gases in the atmosphere is resulting in a warming of the planet. The radiative forcing ($F$, W m$^{-2}$), largely driven by $CO_2$, causes other elements of the climate system to respond to either dampen or amplify the warming. This response is referred to as feedback and quantified by the radiative feedback parameter ($\lambda$, W m$^{-2}$ °C$^{-1}$). It is therefore a

combination of forcings and feedbacks which determine the warming the planet will experience. This can be expressed as $\Delta T$ $= -F / \lambda$. A useful single number proxy for how sensitive the planet is to forcing by $CO_2$ is given by the Equilibrium Climate Sensitivity (ECS, °C). ECS is defined as the temperature rise associated with a doubling of $CO_2$ once the planet has come to equilibrium (which takes more than 1000 years).

Some feedbacks have a relatively low uncertainty. For example, as the planet warms blackbody emissivity increases (Planck

feedback), which dampens warming through a strong negative feedback. However, cloud feedbacks are much more uncertain, exhibiting substantial model-to-model variability (Zelinka et al., 2020; Andrews et al., 2019; Gettelman et al., 2019; Tan et al., 2016). Cloud feedbacks are one of the dominant factors in determining the spread in ECS estimates (Ceppi et al., 2017), and correlate with the cloud feedback parameter (see Figure 1). There are numerous cloud feedbacks which are represented in the overall cloud feedback parameter including feedbacks associated with cloud altitude, cloud amount and cloud albedo.

Of particular relevance for this review is the feedback associated with shallow clouds which exist between 0 and about -35°C in the mid-to high-latitudes. Clouds which contain ice tend to have depleted liquid water paths and therefore lower albedo. Hence, in a warmer world ice will become less prevalent and their albedo will increase; this is the basis of the cloud-phase feedback. There have been significant changes in climate models between CMIP5 and CMIP6, with some models reporting much greater ECS. These higher ECS values are correlated with more positive shallow mid- high-latitude cloud feedbacks in

the CMIP6 models ($r = 0.65$), but only very weakly correlated ($r = 0.17$) in the older CMIP5 models (see Figure 1).

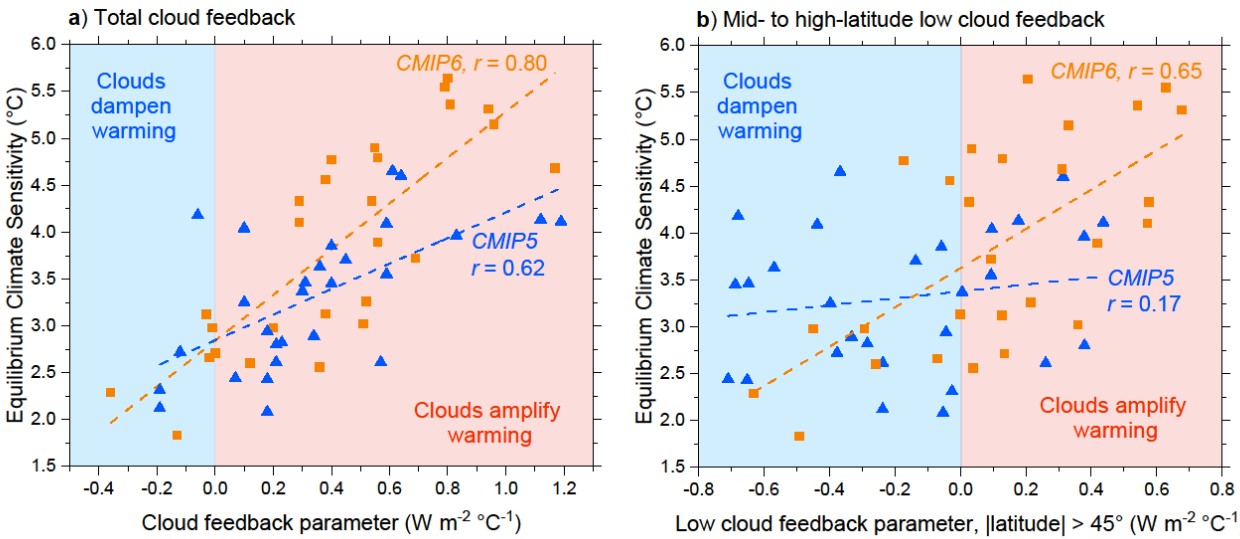

**Figure 1. The equilibrium climate sensitivity plotted against cloud feedback parameter for CMIP5 and CMIP6 models. The left plot is for total cloud feedback parameter, while the right one is for shallow clouds (<680 hPa) which are poleward of 45°. The data is from Zelinka et al. (2020). The correlation between low cloud feedback and ECS which has emerged in CMIP6 models indicates that the treatment of mixed-phase low clouds is critical for driving inter-model ECS variability.**


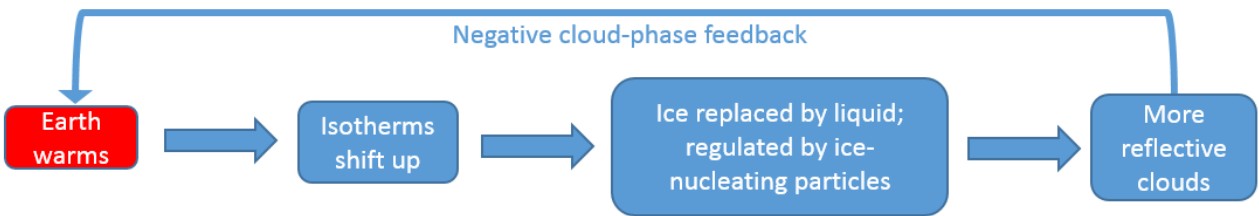

**Figure 2. The cloud-phase feedback and its relationship with ice-nucleating particles (based on Storelvmo et al. (2015)). For shallow marine clouds, the replacement of ice by liquid water leads to more reflective clouds and less shortwave radiation reaching the low**

**albedo ocean surface, resulting in a negative climate feedback.**

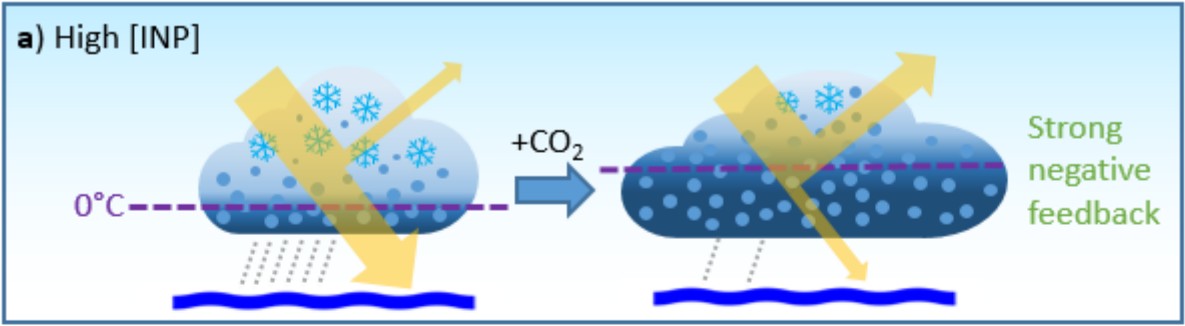

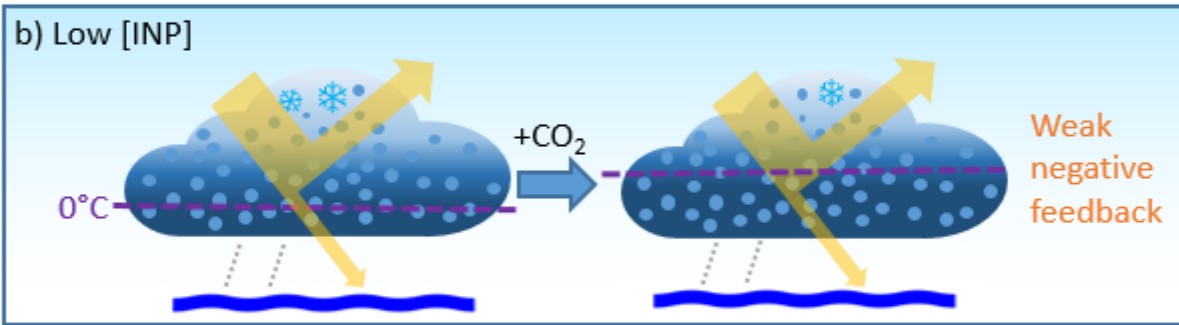

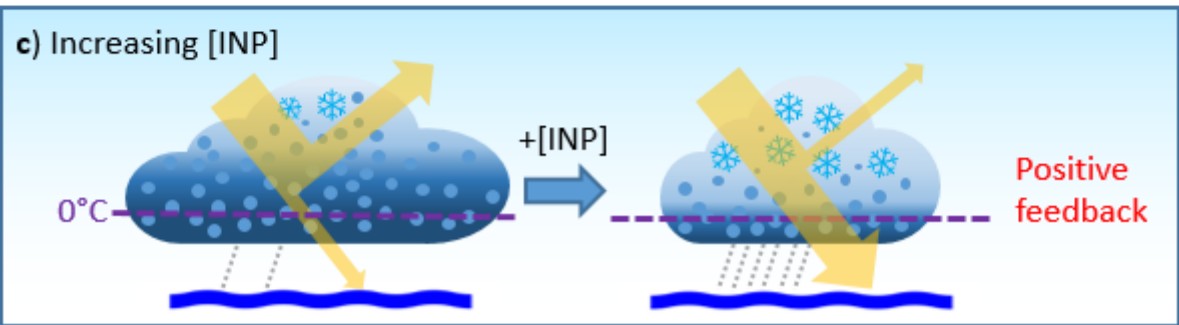

**Figure 3. Cartoons illustrating the response of mixed-phase clouds to a changing climate is controlled by the ice-nucleating particle concentration. a) With a relatively high INP concentration ([INP]), there is a large potential for liquid to replace ice as climate warms and isotherms shift upwards, resulting in a strong negative shortwave feedback. b) With a relatively low INP concentration clouds contain relatively little ice in the present climate, so there is less ice to replace with liquid water and a relatively small negative feedback. c) Setting the temperature changes aside, there may be either increases or decreases in INP concentration in the future which clouds will respond to. We have shown the effect of an increase in INP concentration where we would expect a decrease in liquid water path and a positive feedback.**

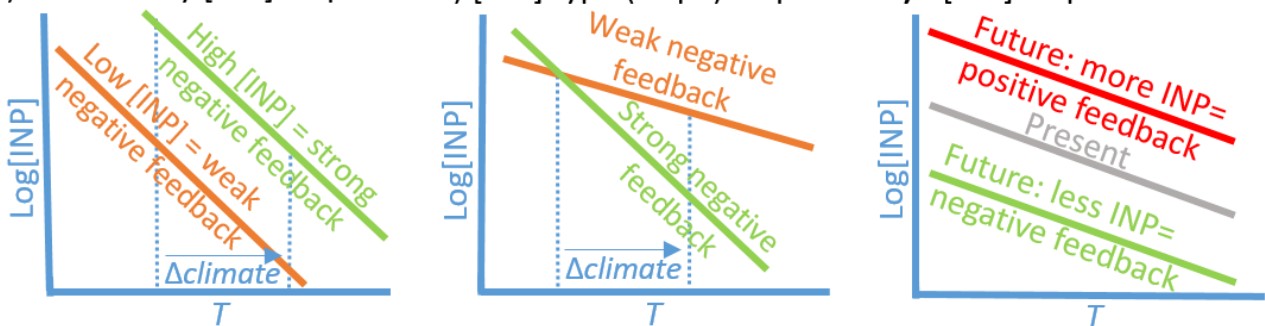

**Figure 4. Illustration of how the cloud-phase feedback depends on the concentration and nature of ice-nucleating particles. The strength of the feedback depends on: a) the present day INP concentration – the cloud phase feedback is more negative with a higher INP concentration since there is a greater potential to replace ice with water on warming (this is mechanistically illustrated in Figure 3a and b); b) the type of INP – different INP types have different temperature dependences (dlog[INP]/d$T$) and for those with steeper temperature dependencies there is a greater potential to replace ice with liquid; c) INP concentrations may change in the future – increases or decreases in INP concentration will feedback on climate positively or negatively, respectively (this relates to Figure 3c).**

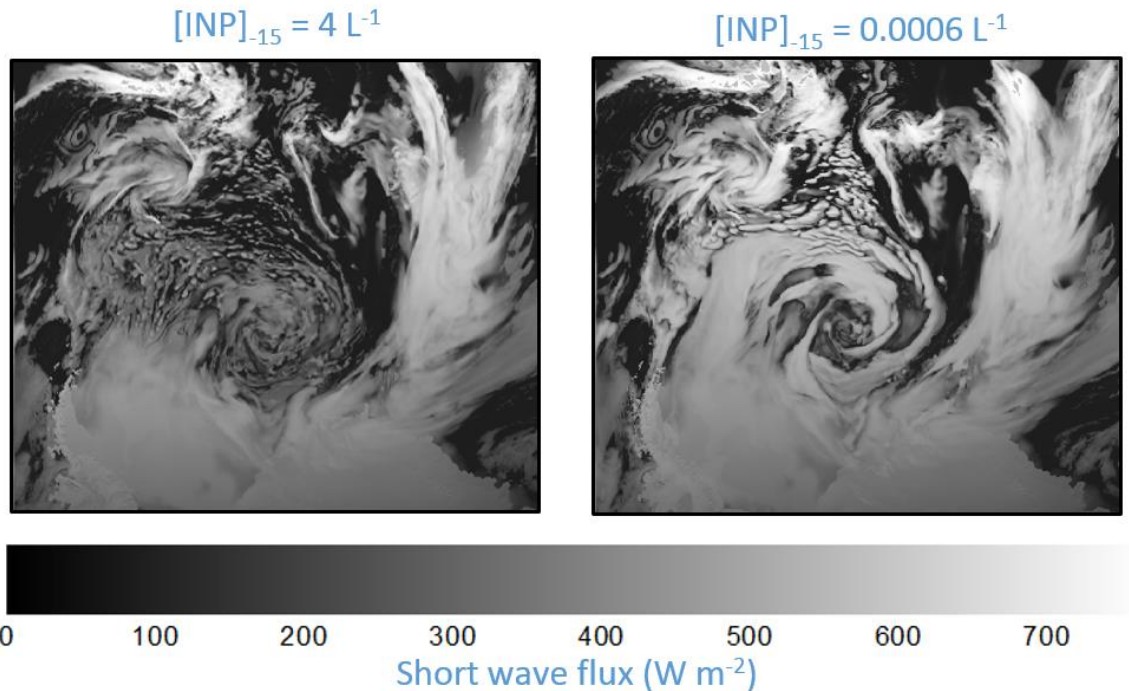

$[INP]_{-15} = 4\ L^{-1}$      $[INP]_{-15} = 0.0006\ L^{-1}$

Short wave flux (W m$^{-2}$)

Figure 5. The effect of INP concentration on model clouds in the cold air sector of a cyclone system over the Southern Ocean with a cloud top temperature of around -15°C (adapted from Vergara-Temprado et al. (2018)). The case is from the 1st March 2015 at 14:00 with 0.07° grid spacings (roughly 7.7 km on a rotated grid). The left map shows a case with a relatively high INP concentration (4 L$^{-1}$ active at -15°C; based on Meyers et al. (1992)) and the right map is for a relatively low INP concentration (0.6 x10$^{-4}$ L$^{-1}$ at -15°C, based on Vergara-Temprado et al. (2017)). The low INP concentration is a good match to measurements in the Southern Ocean (Schmale et al., 2019; McCluskey et al., 2018a) and the higher concentration is within the range of measured INP concentrations elsewhere (see Figure 6). Vergara-Temprado et al. (2018) demonstrate that the lower INP concentration produces clouds which are consistent with satellite measurements of SW flux, whereas the high INP case suffers a large low bias. In the image, the Antarctic peninsula is visible in the lower left and the Antarctic continent is on the bottom right. The x-axis of these plots is approximately 4500 km, while the y-axis is approximately 3900 km.

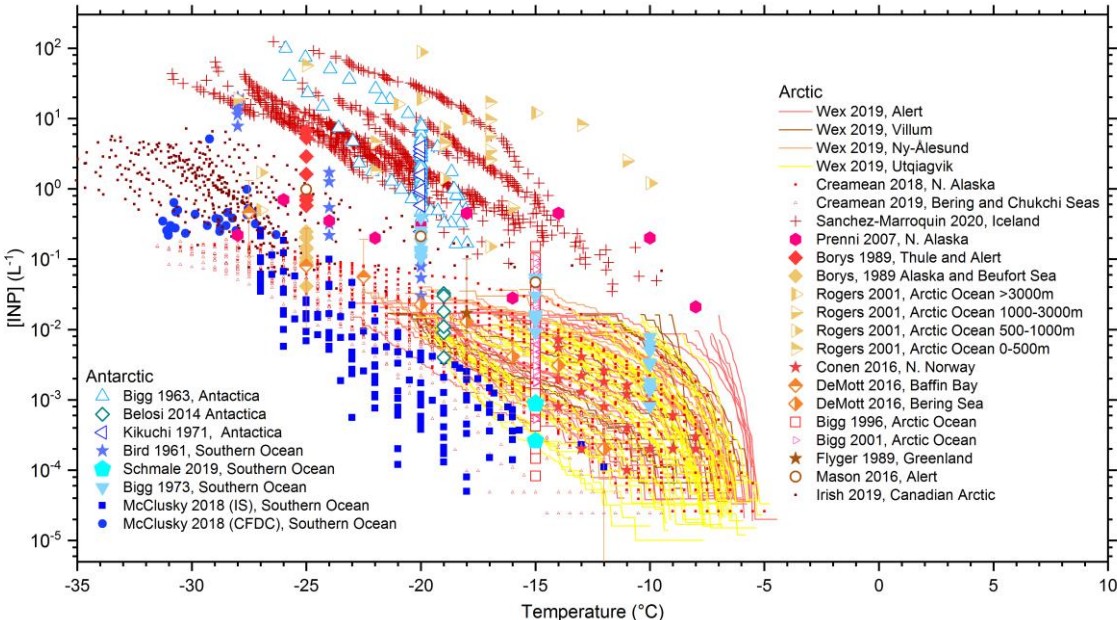

725

**Figure 6. INP concentration measurements in the mid-to high-latitude northern hemisphere (yellows-reds) and the southern hemisphere (blues). Given INP sources in the mid- to high-latitudes are likely to be of central importance to CAOs, we have only presented measurements in either coastal regions or in the open ocean from latitudes greater than about 43°. At present we have no means of predicting this variability in INP concentration, because we are only beginning to quantitatively understand the sources relevant for these regions. Data were taken from multiple sources (Schmale et al., 2019; McCluskey et al., 2018a; Bigg, 1973; Bigg and Hopwood, 1963; Bird et al., 1961; Belosi et al., 2014; Kikuchi, 1971; Sanchez-Marroquin et al., 2020; Wex et al., 2019; Irish et al., 2019a; Creamean et al., 2019; Creamean et al., 2018; Bigg, 1996; Bigg and Leck, 2001; Borys, 1989; Rogers et al., 2001; DeMott et al., 2016; Flyger and Heidam, 1978) and more details are given in Table S1. While there is clearly a great deal of natural variability there are also differences in sampling and instrumentation which will cause some variability. A discussion of known artefacts associated with the technique employed by Bigg and co-workers has been given previously (Mossop and Thorndike, 1966; McCluskey et al., 2018a).**

740