# Peer review of "Opinion: Cloud-phase climate feedback and the importance of ice-nucleating particles"

_Atmospheric Chemistry and Physics, 2020_

## Short Comment (SC1) · 8 Sep 2020

I enjoyed reading this opinion paper. It nicely brings together a number of issues in a call for concerted research to reduce uncertainties regarding feedbacks between climate change, ice-nucleating particles (INPs), and cloud phase and albedo. One issue that I was missing is the effect of rain on the emission of biological INPs (e.g. Bigg & Miles, 1964; Huffman et al., 2013; Hara et al., 2016; Conen et al., 2017; Bigg et al., 2018; Mignani et al., 2020). I thought this issue would be an obvious target for a concerted research effort. Apart from rising temperatures, climate change includes altered precipitation patterns. While, for example, the Mediterranean region is expected

to become dryer, large parts of Siberia will probably experience wetter conditions by the end of this century. Will taking the effect of precipitation on INPs into account not amplify the expected changes in precipitation, with repercussions on model-predicted cloud cover in affected regions?

References

Bigg & Miles, 1964, The results of large-scale measurements of natural ice nuclei, https://doi.org/10.1175/1520-0469(1964)021<0396:TROLMO>2.0.CO;2

Bigg et al., 2018, Persistent after-effects of heavy rain on concentrations of ice nuclei and rainfall suggest a biological cause, www.atmos-chem-phys.net/15/2313/2015/

Conen et al., 2017, Rainfall drives atmospheric ice-nucleating particles in the coastal climate of southern Norway, https://doi.org/10.5194/acp-17-11065-2017

Hara et al., 2016, Variations of ice nuclei concentration induced by rain and snowfall within a local forested site in Japan, https://doi.org/10.1016/j.atmosenv.2015.12.009

Huffman et al., 2013, High concentrations of biological aerosol particles and ice nuclei during and after rain, https://doi.org/10.5194/acp-13-6151-2013

Mignani et al., 2020, Towards parametrising atmospheric concentrations of ice nucleating particles active at moderate supercooling, https://doi.org/10.5194/acp-2020-524

---

## Referee Comment (RC1) · Paul DeMott (Referee) · 10 Sep 2020

**General Comments**

I thank the authors for this stimulating opinion piece. My interest is not only in the topic as a person working in this field, but as someone who also carries a healthy question about how important primary nucleation may be claimed to be in some circumstances, due to observing clouds in many cloud-based field campaigns. Hence, I expected to see secondary ice formation processes brought front and center as a related discussion point, because it may be such an important process in determining cloud properties for radiative transfer and precipitation. To what extent is still unknown

in different cloud scenarios, and to what extent primary ice nucleation processes constrain and feed back to influence secondary processes is also uncertain. I imagine that the thinking here is to provide a special focus on primary nucleation as an independent need, or perhaps also that the higher latitudes are not regions where secondary processes can dominate. The former point is valid, and enough reason to support this opinion piece. I think the latter point is being questioned by some recent studies now making it into the literature. In fact, the complex link between the properties of unfrozen and frozen cloud regions in the same systems is really only alluded to very late in the paper (statement on line 316-317 regarding INP removal processes). With a potentially deeper cloud phase that is $>0°$C and greater cloud depth before freezing ensues, a stimulation of secondary processes and stimulation of warm rain processes could occur. Such an outcome is of course not at all clear, and hence, not only do we need a continued ramping up of data collection and process studies related to primary ice nucleation, but secondary ice formation processes may need just as much attention (perhaps not relegated only to a late mention on lines 322-324 of the piece), and interactions between warm and cold cloud phases (i.e., microphysics in general) deserves attention as a feedback. I am not sure where to put that in the paper, but it could be consolidated. I realize that this paper is attempting to remain finely focused on a key and important variable and objective, and section 4 alludes some other needs while justifying the focus of the opinion. Nevertheless, I wonder if it could be unsatisfying to acquire a bundle of new ice nucleation data, but not have a grasp on being able to properly simulate ice evolution due to lack of understanding of other processes.

This possibly sounds more negative than intended, and odd coming from an ice nucleation researcher. I hope not. Otherwise, I can only support many of the contentions here. There are some truly excellent sections and statements made. I have an assortment of specific comments below, the most critical of which deal with expanding the discussion of some needs. A few are simply editorial.

**Specific Comments**

1) Introduction

Line 58: I looked later in the paper, but did not find this. What will constitute sufficient information about INPs in the regions poleward of 45 degrees? There have been at least two major campaigns in the Southern Ocean region since 2017, the ACE campaign and a suite of studies from 2016 to 2018 that were supported by the U.S. NSF and DOE, and Australian and New Zealand organizations, all including measurements of ice nucleating particles. Some of these are recently published in McCluskey et al. (2018a), Schmale et al. (2019) and Welti et al. (2020), two of which are noted (Welti et al is recently in review in ACP). More measurements are sure to appear. This is not a question of referencing this work though, it is an honest question about the range of spatial and temporal scales that will be needed both in the Southern Ocean and in Arctic regions, where similar campaigns have occurred as referenced in the paper, and more are in the works.

2) The cloud-phase feedback and the importance of ice-nucleating particles

Lines 85-86: While this point about INPs being cloud destroying agents is well-taken, it occurs to me that this paper has mainly considered a uni-directional change in INPs in the future. It could go either way, right? One can imagine either that a warming planet results in increases or decreases in INPs in different regions, and that decreases could be driven by cloud changes as well. That is, the net impact in a remote region like the Southern Ocean is a consequence of gains and losses of INPs, and this is not only affected by source strengths but by scavenging processes.

Lines 93-94: Here one needs to ask if this is a truth or a point for inspection that the INP population controls the amount of ice in most shallow clouds. To some extent this is certainly proven for Arctic clouds, but it is not what has been observed in all clouds over the Southern Ocean, depending on the scales one is referring to. For clouds where secondary ice processes provide the ultimate control on maximum ice concentrations and the distribution of precipitation, this might not be true. A question

is to what extent the secondary process cares about the size of the "trigger" imposed by primary ice formation, and to what extent the areas of secondary processes count from the radiative balance standpoint for wide regions where INP concentrations are generally low. It is difficult for me to support this point strongly, due to the fact that some papers are presently in review and without open access. Nevertheless, I wonder if this point deserves some mention.

Line 95: Do we also need to define the areal extent of critical regions where INP concentrations may be relatively higher or lower? As made apparent in a few recent papers, there are broad regions of the Earth where INPs appear to be well-mixed and relatively uniform (Welti et al., 2020; Schrod et al., 2020), just as there are broad regions like the Southern Ocean where the concentrations are markedly different (reduced) compared to continental regions.

Lines 98-106: I believe there may actually be more to say here. For example, the weaker slope for the fertile soil dust may mask a complete difference in the nature of the INPs versus the mineral dust, speaking of their encompassing both microbial components and their byproduct fragments and organic molecules from their action in soils (e.g., Hill et al., 2016). These other biogenic (primary biological particles and molecular organics), and potentially most important INP sources in the higher temperature regime of supercooled clouds (Garcia et al., 2012; Huffman et al., 2013; O'Sullivan et al., 2018; Mignani et al., 2020; Schneider et al., 2020), may also be altered in a warming future world since they depend on environmental disturbances and conditions at the surface of the Earth. Interestingly, it is clear based on the sheer explosion of recent publication submissions, that the community is already taking up the charge to establish INP spectra and types from different sources, which support the statement on line 105. I think the work beyond the growing number of short and long-term assessments may come in being able to piece out the specific contributors in different source scenarios.

3. To what extent is the persistence of supercooled liquid clouds related to ice nucleation?

Line 146: Missing date on DeMott references. Could be 2016?

Line 150: May I request a definition of biological particles? It may be obvious to the authors, but the wider community reading this may not understand if one means microbes or all biologically-derived INPs (i.e., organic molecules). There is growing evidence that the former are not the same as the latter, in likely following different dependencies (e.g., Mignani et al., 2020; Schneider et al., 2020; Suski et al., 2018). See note below also regarding "biogenic" sources. Also on line 150, why "potentially" combustion particles? There seems ample evidence that biomass burning is a clear source (Schill et al., 2020, and references therein), if not necessarily black carbon.

7. What do we currently know about atmospheric INP in the regions important for the cloud- phase feedback?

Line 215: This is the first use of the term "biogenic" INP sources. It think it is important to be clear on definitions here. It is a point we as a community struggle with still.

Lines 219-220: I hesitate to make this comment, but does Fig. 4 need a qualifier regarding "recognizing that results may to some extent reflect both true INP variability and INP measurement capabilities/uncertainties"? Or "assuming no measurement biases" or "assuming perfect and equivalent measurement capabilities in all studies shown"? Perhaps this is the point of many of your notes in the Supplement, which I only noted late in writing these points.

Section 7.2 INP in the southern mid- to high-latitudes: I wonder if Antarctica is the only consideration as a changing source? There are known land regions impacting the broader region, with variations in transports in different areas that have occurred in the past or may occur in the future (e.g., Neff and Bertler, 2015). In a related regard, the work of Bigg (1973), averaged in Fig. 4, suggests a drastically altered INP scenario now compared that present over 50 years ago. This has been at least briefly discussed

in some of the referenced recent papers, but you make no note of that here.

8. Important areas of future research

Line 281: Should it be "aerosol" generally, rather than "dust"? At least one of the references noted was not specific to dust.

Lines 298-300: Clearly these may be examples familiar to the authors, but you should perhaps mention that other semi-autonomous instrument developments have been occurring for existing technologies within the community (e.g., Bi et al., 2019; Brunner and Kanji, 2020). Others are underway. Also, note that the Möhler et al. (2020) is now in discussion.

**Editorial notes:**

Line 42: This may be a language preference, but I think of higher and lower latitudes, so higher than 45 degrees rather than above 45 degrees. Or perhaps "poleward" of 45 degrees?

Section 6: I am sure that the authors now realize that this section repeats section 5.

Line 345: Should it be Fig. 4 instead of Fig. 3? Also, Jessie's name is misspelled.

**References**

Bi, K., G. R. McMeeking, D. Ding, E. J. T. Levin, P. J. DeMott, D. Zhao, F. Wang, Q. Liu, P. Tian, X. Ma, Y. Chen, M. Huang, H. Zhang, T. Gordon, and P. Chen, 2019: Measurements of ice nucleating particles in Beijing, China. *Journal of Geophysical Research: Atmospheres*, 124, 8065–8075, https://doi.org/10.1029/2019JD030609.

Brunner, C. and Kanji, Z. A.: Continuous online-monitoring of Ice Nucleating Particles: development of the automated Horizontal Ice Nucleation Chamber (HINC-Auto), *Atmos. Meas. Tech. Discuss.*, https://doi.org/10.5194/amt-2020-306, in review, 2020.

Garcia, E., T. C. J. Hill, A. J. Prenni, P. J. DeMott, G. D. Franc and S. M. Kreidenweis,

2012: Biogenic ice nuclei in boundary layer air over two U.S. High Plains agricultural regions, *J. Geophys. Res.*, 117, D18209, doi:10.1029/2012JD018343.

Hill, T. C. J., P. J. DeMott, Y. Tobo, J. Fröhlich-Nowoisky, B. F. Moffett, G. D. Franc, and S. M. Kreidenweis, 2016: Sources of organic ice nucleating particles in soils, *Atmos. Chem. Phys.*, 16, 7195–7211, doi:10.5194/acp-2016-1.

Huffman, J. A., A. J. Prenni, P. J. DeMott, C. Pöhlker, R. H. Mason, N. H. Robinson, J. Fröhlich-Nowoisky, Y. Tobo, V. R. Després, E. Garcia, D. J. Gochis, E. Harris, I. Müller-Germann, C. Ruzene, B. Schmer, B. Sinha, D. A. Day, M. O. Andreae, J. L. Jimenez, M. Gallagher, S. M. Kreidenweis, A. K. Bertram, and U. Pöschl, 2013: High concentrations of biological aerosol particles and ice nuclei during and after rain, *Atmos. Chem. Phys.*, 13, 6151-6164, doi:10.5194/acp-13-6151-2013.

Mignani, C., Wieder, J., Sprenger, M. A., Kanji, Z. A., Henneberger, J., Alewell, C., and Conen, F.: Towards parametrising atmospheric concentrations of ice nucleating particles active at moderate supercooling, *Atmos. Chem. Phys. Discuss.*, https://doi.org/10.5194/acp-2020-524, in review, 2020.

Möhler, O., Adams, M., Lacher, L., Vogel, F., Nadolny, J., Ullrich, R., Boffo, C., Pfeuffer, T., Hobl, A., Weiß, M., Vepuri, H. S. K., Hiranuma, N., and Murray, B. J.: The portable ice nucleation experiment PINE: a new online instrument for laboratory studies and automated long-term field observations of ice-nucleating particles, *Atmos. Meas. Tech. Discuss.*, https://doi.org/10.5194/amt-2020-307, in review, 2020.

Neff, P. D., and Bertler, N. A. N. (2015), Trajectory modeling of modern dust transport to the Southern Ocean and Antarctica, *J. Geophys. Res. Atmos.*, 120, 9303– 9322, doi:10.1002/2015JD023304.

O'Sullivan, D., Adams, M.P., Tarn, M.D. et al. Contributions of biogenic material to the atmospheric ice-nucleating particle population in North Western Europe. *Sci Rep* 8, 13821 (2018). https://doi.org/10.1038/s41598-018-31981-7

Schill, G. P., P. J. DeMott, E. W. Emerson, A. M. C. Rauker, J. K. Kodros, K. J. Suski, T. C. J. Hill, E. J. T. Levin, J. R. Pierce, D. K. Farmer, and S. M. Kreidenweis. The contribution of black carbon to global ice nucleating particle concentrations relevant to mixed-phase clouds. *Proceedings of the National Academy of Sciences*, 2020; 202001674 DOI: 10.1073/pnas.2001674117

Schneider, J., Höhler, K., Heikkilä, P., Keskinen, J., Bertozzi, B., Bogert, P., Schorr, T., Umo, N. S., Vogel, F., Brasseur, Z., Wu, Y., Hakala, S., Duplissy, J., Moisseev, D., Kulmala, M., Adams, M. P., Murray, B. J., Korhonen, K., Hao, L., Thomson, E. S., Castarède, D., Leisner, T., Petäjä, T., and Möhler, O.: The seasonal cycle of ice-nucleating particles linked to the abundance of biogenic aerosol in boreal forests, *Atmos. Chem. Phys. Discuss.*, https://doi.org/10.5194/acp-2020-683, in review, 2020.

Schrod, J., Thomson, E. S., Weber, D., Kossmann, J., Pöhlker, C., Saturno, J., Ditas, F., Artaxo, P., Clouard, V., Saurel, J.-M., Ebert, M., Curtius, J., and Bingemer, H. G.: Long-term INP measurements from four stations across the globe, *Atmos. Chem. Phys. Discuss.*, https://doi.org/10.5194/acp-2020-667, in review, 2020.

Suski, K. J., T. C. J. Hill, E. J. T. Levin, A. Miller, P. J. DeMott, and S. M. Kreidenweis, 2018: Agricultural harvesting emissions of ice-nucleating particles, *Atmos. Chem. Phys.*, 18, 13755-13771, https://doi.org/10.5194/acp-18-13755-2018.

Welti, A., Bigg, E. K., DeMott, P. J., Gong, X., Hartmann, M., Harvey, M., Henning, S., Herenz, P., Hill, T. C. J., Hornblow, B., Leck, C., Löffler, M., McCluskey, C. S., Rauker, A. M., Schmale, J., Tatzelt, C., van Pinxteren, M., and Stratmann, F.: Ship-based measurements of ice nuclei concentrations over the Arctic, Atlantic, Pacific and Southern Ocean, *Atmos. Chem. Phys. Discuss.*, https://doi.org/10.5194/acp-2020-466, in review, 2020.

---

## Referee Comment (RC2) · Trude Storelvmo (Referee) · 16 Oct 2020

I'd like to congratulate the authors on an important and well written opinion article, and generally agree with the main findings and recommendations. A few things that could be worth adding in a revised manuscript are: i) While INPs are important, there is a general lack of understanding also of the other (subsequent) processes governing cloud glaciation (secondary ice production, WBF process, riming, seeder-feider, etc). These processes tend to matter way more than INPs when it comes to cloud phase in GCMs. In other words, even with perfect knowledge of INPs, a better cloud phase feedback representation is not guaranteed. This should be stressed more. ii) The idea

that INPs could increase in abundance in future in response to warming is intriguing, but not supported by paleoclimate records in which cold=dusty and warm=dust-free. This should be acknowledged. iii) the paper is generally well written, but fixing a few typos towards the end of the paper would make it even better.

---

## Author Comment (AC1) · 7 Dec 2020

We thank Franz Conen for his comment on the role of precipitation releasing INP to the atmosphere.

We have added the following statement to section 2: "Also, INP emissions have been linked to environmental factors such as rain fall, hence a warmer wetter world may lead to enhanced INP emission rates from some terrestrial sources (Conen et al., 2017; Huffman et al., 2014; Hara et al., 2016)."

[Figure]

2020.

---

## Author Comment (AC2) · 7 Dec 2020

**Response to Paul DeMott's comments**

We thank Paul DeMott for his comments which we have addressed point by point:

General Comments: We have not reproduced the whole of Paul DeMott's text here, but the main thrust of his point is that we have not placed sufficient focus on other microphysical processes that are important (in particular secondary ice production). DeMott summarised this in the final few lines: '*Nevertheless, I wonder if it could be unsatisfying to acquire a bundle of new ice nucleation data, but not have a grasp on being able to properly simulate ice evolution due to lack of understanding of other processes. This possibly sounds more negative than intended, and odd coming from an ice nucleation researcher. I hope not. Otherwise, I can only support many of the contentions here. There are some truly excellent sections and statements made.*'  We note that Storelvmo also made a similar comment.

We agree that other microphysical processes are also important and did not intend to imply they are not. However, we also think that primary ice nucleation needs special focus and is one of, if not the, least well understood process. As such it is our hypothesis that our understanding of this process limits the accuracy of our models.  But, yes, once we get to the point of better defining INP and primary production, uncertainty in other processes will become limiting and we must not forget about these processes.

We have reorganised the pertinent paragraph in section 2 into two paragraphs and made it clearer that other processes are also important. As part of this reorganisation, we make the following statements which directly address the referee's comments:

"The shift to fewer, but larger hydrometers when a supercooled cloud glaciates is a result of the abundance of aerosol available for nucleating cloud droplets and ice crystals, as well as the various ice-related microphysical processes which occur subsequent to ice nucleation."; "In some situations the impact of INP will be amplified through secondary ice production (SIP) where a range of mechanisms are thought to result in the production of additional ice crystals (Field et al., 2017). It should be borne in mind that these processes (SIP, WBF, riming) subsequent to ice nucleation are also relatively poorly understood and also need attention (Komurcu et al., 2014). However, primary ice production initiates these subsequent ice-related processes, therefore the role of INPs in the cloud-phase feedback is the focus of this paper"; "However, the relationship between INP concentration and cloud glaciation is complex and governed by the WBF process (Desai et al., 2019)."

At the end of section 2 we now state:

"The fact that ECS is sensitive to the balance between supercooled water and ice in clouds means that we have to improve our understanding of ice-related microphysical processes. In particular, we need a concerted effort to understand the atmospheric abundance of INPs, the aerosol type which catalyses ice formation in mixed phase clouds and plays a major role in defining the cloud-phase feedback."

At the end of section 4 we now state:

"In the future, models need to improve their representation of ice-related microphysical processes, in particular, they need to include a direct link to aerosol type, specifically INP, in order to improve the representation of clouds phase and the response of clouds to a warming world."

Citations:

We have added the new citations mentioned by DeMott in the appropriate places.  We have made the decision not to include data from papers in review in ACPD in our data compilation, but have cited these papers for their general conclusions throughout the paper.

Specific Comments C2

*1) Line 58: I looked later in the paper, but did not find this. What will constitute sufficient information about INPs in the regions poleward of 45 degrees? There have been at least two major campaigns in the Southern Ocean region since 2017, the ACE campaign and a suite of studies from 2016 to 2018 that were supported by the U.S. NSF and DOE, and Australian and New Zealand organizations, all including measurements of ice nucleating particles. Some of these are recently published in McCluskey et al. (2018a), Schmale et al. (2019) and Welti et al. (2020), two of which are noted (Welti et al is recently in review in ACP). More measurements are sure to appear. This is not a question of referencing this work though, it is an honest question about the range of spatial and temporal scales that will be needed both in the Southern Ocean and in Arctic regions, where similar campaigns have occurred as referenced in the paper, and more are in the works.*

Our statement was rather vague. It was meant as an indicator of the structure of the paper.  The campaign mentioned by the referee are included in Fig 4 (with the exception of the Welti paper, which is in review).  We have been more specific in a revised sentence: "While we have learnt a great deal from recent field and laboratory work about INPs in mid- to high-latitudes (~45-70°), the region critical for the cloud-phase feedback, we need a much better understanding of sources and sinks of INP as well as the nature of INPs in both hemispheres."

2) *Lines 85-86: While this point about INPs being cloud destroying agents is well-taken, it occurs to me that this paper has mainly considered a uni-directional change in INPs in the future. It could go either way, right? One can imagine either that a warming planet results in increases or decreases in INPs in different regions, and that decreases could be driven by cloud changes as well. That is, the net impact in a remote region like the Southern Ocean is a consequence of gains and losses of INPs, and this is not only affected by source strengths but by scavenging processes.*

This is correct.  We have edited the last line in the abstract to read: "We also need to develop a predictive capability for future INP emissions **and sinks** in a warmer world….."  We do already introduce the idea that INP in the future might increase or decrease in Figure 2, but have clarified this in the text.  In section 2 we now state: "Thirdly, INP sources, processing and removal in the atmosphere are also likely to change with a changing climate.", and "Alternatively, loss mechanisms might be enhanced in a warmer world with more precipitation"

*3) Lines 93-94: Here one needs to ask if this is a truth or a point for inspection that the INP population controls the amount of ice in most shallow clouds. To some extent this is certainly proven for Arctic clouds, but it is not what has been observed in all clouds over the Southern Ocean, depending on the scales one is referring to. For clouds where secondary ice processes provide the ultimate control on maximum ice concentrations and the distribution of precipitation, this might not be true. A question is to what extent the secondary process cares about the size of the "trigger" imposed by primary ice formation, and to what extent the areas of secondary processes count from the radiative balance standpoint for wide regions where INP concentrations are generally low. It is difficult for me to support this point strongly, due to the fact that some papers are presently in review and without open access. Nevertheless, I wonder if this point deserves some mention*.

We have expanded section 2 to include more on secondary production (see response above).  Also, we have edited the pertinent line removing the words 'primarily influenced' with 'strongly

influenced': "Since the amount of ice in many shallow clouds is strongly influenced by the INP population, there are likely to be regional and seasonal variations in the cloud-phase feedback."

*4) Line 95: Do we also need to define the areal extent of critical regions where INP concentrations may be relatively higher or lower? As made apparent in a few recent papers, there are broad regions of the Earth where INPs appear to be well-mixed and relatively uniform (Welti et al., 2020; Schrod et al., 2020), just as there are broad regions like the Southern Ocean where the concentrations are markedly different (reduced) compared to continental regions.*

This would be an interesting exercise. We note that Welti et al. has made a good start in defining the areal extent of INP concentrations. As part of the M-Phase project we are working on doing exactly this with a standardised database of INP measurements.

*5) Lines 98-106: I believe there may actually be more to say here. For example, the weaker slope for the fertile soil dust may mask a complete difference in the nature of the INPs versus the mineral dust, speaking of their encompassing both microbial components and their byproduct fragments and organic molecules from their action in soils (e.g., Hill et al., 2016). These other biogenic (primary biological particles and molecular organics), and potentially most important INP sources in the higher temperature regime of supercooled clouds (Garcia et al., 2012; Huffman et al., 2013; O'Sullivan et al., 2018; Mignani et al., 2020; Schneider et al., 2020), may also be altered in a warming future world since they depend on environmental disturbances and conditions at the surface of the Earth. Interestingly, it is clear based on the sheer explosion of recent publication submissions, that the community is already taking up the charge to establish INP spectra and types from different sources, which support the statement on line 105. I think the work beyond the growing number of short and long-term assessments may come in being able to piece out the specific contributors in different source scenarios.*

There certainly is a lot more that could be said, but we tried to break this section down into key points and keep it brief. This point of changing INP with changing climate fits better into the third point. We have inserted: "Furthermore, biological processes which result in very active biogenic INP (primary biological particles, by-product fragments and macromolecules) (Hill et al., 2016; O'Sullivan et al., 2015), may also respond to a changing climate."

*6) Line 146: Missing date on DeMott references. Could be 2016?*

Corrected.

*7) Line 150: May I request a definition of biological particles? It may be obvious to the authors, but the wider community reading this may not understand if one means microbes or all biologically-derived INPs (i.e., organic molecules). There is growing evidence that the former are not the same as the latter, in likely following different dependencies (e.g., Mignani et al., 2020; Schneider et al., 2020; Suski et al., 2018). See note below also regarding "biogenic" sources. Also on line 150, why "potentially" combustion particles? There seems ample evidence that biomass burning is a clear source (Schill et al., 2020, and references therein), if not necessarily black carbon.*

We now have a definition for biogenic in the first use of the work in section 2 (see point 5 above). We have removed 'potentially' from the reference to combustion aerosol.

*8) Line 215: This is the first use of the term "biogenic" INP sources. It think it is important to be clear on definitions here. It is a point we as a community struggle with still.*

*See new definition in point 5.*

*9) Lines 219-220: I hesitate to make this comment, but does Fig. 4 need a qualifier regarding "recognizing that results may to some extent reflect both true INP variability and INP measurement capabilities/uncertainties"? Or "assuming no measurement biases" or "assuming perfect and equivalent measurement capabilities in all studies shown"? Perhaps this is the point of many of your notes in the Supplement, which I only noted late in writing these points.*

We did signpost the reader to literature where these issue are discussed in the caption where we state : "A discussion of known artefacts associated with some older techniques has been given previously (Mossop and Thorndike, 1966; McCluskey et al., 2018).".  We have added to this to make this clearer: "While there is clearly a great deal of natural variability there are also differences in sampling and instrumentation which will cause some variability. A discussion of known artefacts associated with techniques has been given previously (Mossop and Thorndike, 1966; McCluskey et al., 2018a)."

*10) Section 7.2 INP in the southern mid- to high-latitudes: I wonder if Antarctica is the only consideration as a changing source? There are known land regions impacting the broader region, with variations in transports in different areas that have occurred in the past or may occur in the future (e.g., Neff and Bertler, 2015).*

We have added the following statement: "There are also dust sources more generally across the southern hemisphere, in particular dust from New Zealand and Patagonia are transported to the higher latitude Southern Ocean (Neff and Bertler, 2015) and dust from Patagonia has been shown to be effective at nucleating ice (López et al., 2018)."

*11) In a related regard, the work of Bigg (1973), averaged in Fig. 4, suggests a drastically altered INP scenario now compared that present over 50 years ago. This has been at least briefly discussed in some of the referenced recent papers, but you make no note of that here.*

In one draft we had an extensive discussion of this, but removed it because we wanted to keep the discussion brief and focused on the importance of INP for climate rather than have lengthy discussions of potential measurement issues. At face value the measurements imply that INP concentrations have changed over time, however this conclusion has to be set against what is known about the technique Bigg employed. There is a documented dependence of the apparent INP concentration on the amount of air sampled, which indicates that there is a fundamental problem with the technique employed (see citations in the caption of Fig 4).  We also refer to this in the SI table.  We would rather not get into this complex issue in the main body of the paper.

*12) Line 281: Should it be "aerosol" generally, rather than "dust"? At least one of the references noted was not specific to dust.*

Corrected.

*Lines 298-300: Clearly these may be examples familiar to the authors, but you should perhaps mention that other semi-autonomous instrument developments have been occurring for existing technologies within the community (e.g., Bi et al., 2019; Brunner and Kanji, 2020). Others are underway. Also, note that the Möhler et al. (2020) is now in discussion.*

Citations added.

*Editorial notes: Line 42: This may be a language preference, but I think of higher and lower latitudes, so higher than 45 degrees rather than above 45 degrees. Or perhaps "poleward" of 45 degrees?*

Changed to "poleward of 45" here and in Figure 1 caption.

*Section 6: I am sure that the authors now realize that this section repeats section 5.*

We have corrected this embarrassing error.

*Line 345: Should it be Fig. 4 instead of Fig. 3? Also, Jessie's name is misspelled.*

Corrected

**References**

Hill, T. C. J., DeMott, P. J., Tobo, Y., Froehlich-Nowoisky, J., Moffett, B. F., Franc, G. D., and Kreidenweis, S. M.: Sources of organic ice nucleating particles in soils, Atmos. Chem. Phys., 16, 7195-7211, 10.5194/acp-16-7195-2016, 2016.

Komurcu, M., Storelvmo, T., Tan, I., Lohmann, U., Yun, Y., Penner, J. E., Wang, Y., Liu, X., and Takemura, T.: Intercomparison of the cloud water phase among global climate models, J. Geophys. Res., 119, 3372-3400, 10.1002/2013JD021119, 2014.

López, M. L., Borgnino, L., and Ávila, E. E.: The role of natural mineral particles collected at one site in Patagonia as immersion freezing ice nuclei, Atmos. Res., 204, 94-101, https://doi.org/10.1016/j.atmosres.2018.01.013, 2018.

McCluskey, C. S., Hill, T. C. J., Humphries, R. S., Rauker, A. M., Moreau, S., Strutton, P. G., Chambers, S. D., Williams, A. G., McRobert, I., Ward, J., Keywood, M. D., Harnwell, J., Ponsonby, W., Loh, Z. M., Krummel, P. B., Protat, A., Kreidenweis, S. M., and DeMott, P. J.: Observations of Ice Nucleating Particles Over Southern Ocean Waters, Geophys. Res. Lett., 45, 11,989-911,997, 10.1029/2018gl079981, 2018.

Mossop, S. C., and Thorndike, N. S. C.: The Use of Membrane Filters in Measurements of Ice Nucleus Concentration. I. Effect of Sampled Air Volume, J. App. Meteorol., 5, 474-480, 10.1175/1520-0450(1966)005<0474:TUOMFI>2.0.CO;2, 1966.

Neff, P. D., and Bertler, N. A. N.: Trajectory modeling of modern dust transport to the Southern Ocean and Antarctica, J. Geophys. Res., 120, 9303-9322, https://doi.org/10.1002/2015JD023304, 2015.

O'Sullivan, D., Murray, B. J., Ross, J. F., Whale, T. F., Price, H. C., Atkinson, J. D., Umo, N. S., and Webb, M. E.: The relevance of nanoscale biological fragments for ice nucleation in clouds, Scientific Reports, 5, 10.1038/srep08082, 2015.

---

## Author Comment (AC3) · 7 Dec 2020

**Response to Trude Storelvmo's review**

We thank Trude Storelvmo for her comments which we have addressed point by point:

*I'd like to congratulate the authors on an important and well written opinion article, and generally agree with the main findings and recommendations. A few things that could be worth adding in a revised manuscript are:*

*i) While INPs are important, there is a general lack of understanding also of the other (subsequent) processes governing cloud glaciation (secondary ice production, WBF process, riming, seeder-feeder, etc). These processes tend to matter way more than INPs when it comes to cloud phase in GCMs. In other words, even with perfect knowledge of INPs, a better cloud phase feedback representation is not guaranteed. This should be stressed more.*

A similar point was also raised by Paul DeMott. We have made significant changes to the manuscript to make it clear that other processes are also important. Please see our response to DeMott for details.

*ii) The idea that INPs could increase in abundance in future in response to warming is intriguing, but not supported by paleoclimate records in which cold=dusty and warm=dust-free. This should be acknowledged.*

This is an interesting point. However, we stress that many INP at the mid-high latitudes may not be dust. In section 6.1 where we discuss this we refer to Wex et al. (2019) who find that that the biological INP active at the warmest temperatures increase in concentration in snow free periods. But, in addition, what will happen to the dust sources of most relevance to CAOs in the future (and their relation to ice and sediment cores) is also not clear. We have added the following brief discussion on what might happen to dust sources in a future world: "In addition to this, it has been argued that high latitude dust sources associated with glaciers will become more active in the future (Bullard et al., 2016) and it was recently shown that mineral dust emissions from the coastal areas of Greenland have increased in the last few decades (Amino et al., 2020). However, in contrast paleo records indicate that warmer periods are generally less dusty than dry periods, although this may reflect a combination of lower latitude sources being more active and increased transport to high latitudes during glacial periods (Lamy et al., 2014; Fischer et al., 2007). Hence, it may be that the glacial-interglacial trends in high latitude dust sources relevant for CAOs are decoupled from low latitude dust sources. More work in identifying the sources of INP in the high latitudes and how these sources will respond to a changing climate is clearly required."

We have added a link to section 6 in section 2 where we introduce the idea of INP changes with climate as our third hypothesis.

We have also adjusted the text to make it clear that it is not only existing sources that are likely to increase in emission strength, but more sources may become available. In section 2 "For example, it has been suggested that less snow and ice cover may lead to more widespread emission sources and higher dust emissions rates at high latitudes " and section 6.1: "…these sources may be active for more of the year and more sources may become available…".

*iii) the paper is generally well written, but fixing a few typos towards the end of the paper would make it even better.*

We have corrected the replication of section 5 and proof read the manuscript.

**References**

Amino, T., Iizuka, Y., Matoba, S., Shimada, R., Oshima, N., Suzuki, T., Ando, T., Aoki, T., and Fujita, K.: Increasing dust emission from ice free terrain in southeastern Greenland since 2000, Polar Science, 100599, https://doi.org/10.1016/j.polar.2020.100599, 2020.

Bullard, J. E., Baddock, M., Bradwell, T., Crusius, J., Darlington, E., Gaiero, D., Gassó, S., Gisladottir, G., Hodgkins, R., McCulloch, R., McKenna-Neuman, C., Mockford, T., Stewart, H., and Thorsteinsson, T.: High-latitude dust in the Earth system, Rev. Geophys., 54, 447-485, 10.1002/2016RG000518, 2016.

Fischer, H., Siggaard-Andersen, M.-L., Ruth, U., Röthlisberger, R., and Wolff, E.: Glacial/interglacial changes in mineral dust and sea-salt records in polar ice cores: Sources, transport, and deposition, Rev. Geophys., 45, https://doi.org/10.1029/2005RG000192, 2007.

Lamy, F., Gersonde, R., Winckler, G., Esper, O., Jaeschke, A., Kuhn, G., Ullermann, J., Martinez-Garcia, A., Lambert, F., and Kilian, R.: Increased Dust Deposition in the Pacific Southern Ocean During Glacial Periods, Science, 343, 403-407, 10.1126/science.1245424, 2014.

---

## Author Response (AR2)

Dear Professor Koop

Thank you for your comments on our manuscript. We have made the necessary changes in the attached document and addressed your points below:

*Lines 98-130: Apparently, there is no reference to Fig. 2e) in the main text (at least I did not spot one).*

    Corrected

*Line 174: "it has been show that" -> "it has been shown that"*

    Corrected

*Line 203: I suggest adding a comma after: "… impacted by INP"*

    Corrected

*Lines 203-204: "low INP leading" -> "low INP concentrations leading"; "high INP tends" -> "high INP concentrations tend"*

    Corrected

*Figure 1: Please provide figure in vector format.*

    *We have remade this figure to make this possible and will upload as a .pdf*

*Figure 2: I found myself struggling to understand Figure 2, in particular as the color coding suggested apparent relations, which I found (after carefully reading the text) were actually not intended. Moreover, the overall arrangement of the figure seems to imply relations between panel b) and e), c) and f), and d) and g). However, panel b) and c) are both related to panel e), panel d) is related to panel g), and panel f) is not directly related to any of the panels b) to d). An improvement of the color coding and maybe other changes such as a different arrangement may help the reader to capture the interconnections immediately. I deem this important, as Figure 2 is probably the most important figure of the paper.*

    We have decided to separate these this panel into three separate figures. The combination in one figure was causing some confusion since the arrangement wrongly implied relationships which were not intended. Fig 2 is now the block diagram, fig 3 are the three cartoons and fig 4 contains the three sketch plots. We have also updated the text accordingly.

*1. The horizontal green dashed lines in panels b) to d) are unrelated to the green lines in panels e) to g). Hence, I suggest a different color for the horizontal lines.*

    We changed this colour to purple.

*2. Initially I was wondering, to what the color coding of the feedback statements in panels b) to d) relate to? Green and orange*
*appears in almost every panel e) to g). However, apparently, both panels b) and c) relate to the colored lines in panel e).*
*Maybe this connection can be made more obvious? This may be achieved by giving each line in panels f) and g) a distinctive*
*unmistakable color. For example, the two lines in panel f) are not related to panels b) to d) and, hence, can have entirely*
*different colors. And those in panel g) yet other colors with that on the right hand side (Future: more INP) given the same as*
*the feedback statement in panel d).*

We have improved the colour coding.  Red corresponds to positive forcing, while green is for strong negative forcing and orange is for weaker negative forcing.

3. I was wondering whether in panel e) the green line should be above rather than to the right of the orange one. Then you would be able to add the same blue delta climate arrow as in f) and the effect can be seen immediately.

Yes, we see the advantage of this and have made the change.

[revised manuscript text omitted]